# Efficient qudit based scheme for photonic quantum computing

M. Karácsony[1*], L. Oroszlány[1,2] and Z. Zimborás[3,4]

**1** Department of Physics of Complex Systems, ELTE Eötvös Loránd University, 1117 Budapest, Hungary
**2** MTA-BME Lendület Topology and Correlation Research Group, Budapest University of Technology and Economics, 1521 Budapest, Hungary
**3** Wigner Research Centre for Physics, H-1525, P.O.Box 49, Budapest, Hungary
**4** Algorithmiq Ltd, Kanavakatu 3C, FI-00160 Helsinki, Finland
* karacsm@student.elte.hu

April 12, 2023

## Abstract

**Linear optics is a promising alternative for the realization of quantum computation protocols due to the recent advancements in integrated photonic technology. In this context usually qubit based quantum circuits are considered, however, photonic systems naturally allow also for $d$-ary, i.e., qudit based, algorithms. This work investigates qudits defined by the possible photon number states of a single photon in $d > 2$ optical modes. We demonstrate how to construct locally optimal non-deterministic many-qudit gates using linear optics and photon number resolving detectors, and explore the use of qudit cluster states in the context of a $d$-ary optimization problem. We find that the qudit cluster states require less optical modes and are encoded by a fewer number of entangled photons than the qubit cluster states with similar computational capabilities. We illustrate the benefit of our qudit scheme by applying it to the k-coloring problem.**

# 1 Introduction

In recent years, many advancements have been made in the area of integrated photonic quantum technologies. Integrated quantum photonic processors (IQP) with on-chip linear optics [1,2], single photon sources [3–5] and photon number resolving detectors (PNRD) [6–8] have been demonstrated. These integrated components and waveguides are usually printed onto silica, which makes it possible to miniaturize the chips and increase the number of on-chip components. Since photons couple very weakly to their environment, IQPs typically do not require millikelvin temperatures to operate as opposed to other quantum information processing architectures like superconducting devices [9]. However, there are still many hurdles to overcome in order to achieve fully scalable and fault-tolerant quantum computation. Until then even IQPs will remain intermediate scaled and thus the amount of available quantum resources will be limited. A common technique for reducing computational resources is to encode quantum information efficiently. Typically quantum information is encoded into two level systems, qubits which became standard due to their simplicity and analogy to classical computation. However, there are other ways to deal with quantum information. An alternative is to use qudits, $d > 2$ level quantum systems, which are the basis for high-dimensional quantum computation [10]. Qudits have been considered in many areas of quantum computation, for example in topological quantum computation the braiding of $\mathbb{Z}_d$ parafermions [11] provide a natural way to implement qudits [12]. There also have been attempts to realize superconducting qudits [13,14], but because of the flexibility in how photon states can be interpreted, qudits are most prominent in photonics [15–20]. The use of qudits increases the number of dimensions per computation unit. This property of qudits in theory can provide some benefits over qubit systems by reducing circuit complexity of quantum algorithms or by increasing the channel capacity and noise tolerance of communication protocols [21,22].

There has been considerable experimental progress on the way towards photonic quantum supremacy via boson sampling (see, for example, Refs. [23–25]), however, the realization of universal photonic quantum computation still seems far from reality. This is because linear optics alone cannot be used to produce certain multi-photon states which are required in some form by all universal photonic schemes. The most well-known universal photonic scheme is the KLM scheme [26] which solves the problem of creating entanglement between photonic qubits by applying postselection based on ancilla measurements. This makes it possible to prepare entangled two qubit states non-deterministically using only linear optics and PNRDs. This measurement based method combined with the quantum teleportation of qubits was shown to be able to create near-deterministic two qubit gates. However, the complete implementation of the KLM scheme proved to be out of reach with current technology, and only some of the its techniques has been demonstrated experimentally. Another approach for realizing universal quantum computation is to use measurement based quantum computation (MBQC). In MBQC the quantum algorithms are performed by making single qubit measurements on a large entangled state, usually a cluster state [27,28]. For photonics MBQC provides a resource efficient and deterministic way to implement universal quantum computation. Since using qudits instead of qubits are often beneficial in quantum computation, as discussed above, it is natural to ask whether photonic MBQC can also be improved by utilizing high-dimensional computational units. In this work, we show that qudit cluster states can indeed be more resource efficient than qubit cluster states.

The present paper is structured as follows: In Sec. 2 we go over a high-dimensional generalization of the KLM encoding and explain how to use passive linear optics to create single- and many-qudit gates. We generalize the post selection based approach of the KLM scheme to implement non-deterministic many-qudit gates. Then we use an optimization method to find locally optimal interferometer configurations for some two-qudit gates. In Sec. 3 we intro-

duce the theory of high-dimensional clusters states, and then in Sec. 4 we show how to use high-dimensional cluster states to solve the $k$-coloring problem.

## 2 Multi-rail qudit encoding

In this section, we describe a simple method for encoding d-level quantum system, i.e., qudits, into optical modes and show how to implement non-deterministic many-qudit gates using linear optics and PNRDs. Before describing the qudit encoding, we present a few basic qudit gates and discuss some useful identities between them. Then, after the specific description of the multi-rail encoding, we also present an optimization method for increasing the success rate of non-deterministic photonic many-qudit gates.

### 2.1 Qudit gates

There are several different conventions in use for the generalization of the well known Pauli $X$ and $Z$ gates for qudit systems. We will use the following definitions

$$X = \sum_{n=0}^{d-1} |n \oplus 1\rangle \langle n|, \qquad Z = \sum_{n=0}^{d-1} \omega^n |n\rangle \langle n|, \tag{1}$$

where $\oplus$ means addition modulo $d$ and $\omega = \exp(i2\pi/d)$. One can immediately see that $X$ and $Z$ are unitary and $X^d = Z^d = \mathbb{1}$. However, it is important to note that they are not Hermitian for $d > 2$. Since the qudit gates $X$ and $Z$ defined above have the same eigenvalues, they satisfy a similarity relation,

$$HXH^\dagger = Z, \tag{2}$$

where the corresponding basis change operator defines the qudit Hadamard gate

$$H = \frac{1}{\sqrt{d}} \sum_{n=0}^{d-1} \sum_{m=0}^{d-1} \omega^{nm} |n\rangle \langle m|. \tag{3}$$

The Hadamard gate is often used to prepare the qudit state

$$|+\rangle \equiv H|0\rangle = \frac{1}{\sqrt{d}} \sum_{n=0}^{d-1} |n\rangle, \tag{4}$$

which plays a key role in many quantum algorithms. The controlled $X$ and $Z$ gates are defined as

$$CX = \sum_{n=0}^{d-1} |n\rangle \langle n| \otimes X^n, \qquad CZ = \sum_{n=0}^{d-1} |n\rangle \langle n| \otimes Z^n. \tag{5}$$

From Eq. (2) it follows that

$$CX = \left(\mathbb{1} \otimes H^\dagger\right) CZ \left(\mathbb{1} \otimes H\right). \tag{6}$$

Equation (5) means that $CX|n\rangle \otimes |m\rangle = |n\rangle \otimes |n \oplus m\rangle$, thus one can write

$$\begin{aligned}
CX^\dagger(\mathbb{1} \otimes Z)CX|n\rangle \otimes |m\rangle &= CX^\dagger(\mathbb{1} \otimes Z)|n\rangle \otimes |n \oplus m\rangle \\
&= CX^\dagger \omega^{n+m} |n\rangle \otimes |n \oplus m\rangle \\
&= \omega^{n+m} |n\rangle \otimes |m\rangle = Z \otimes Z|n\rangle \otimes |m\rangle,
\end{aligned} \tag{7}$$

where $n, m \in \{0, 1, 2, \ldots d-1\}$, therefore

$$CX^\dagger(\mathbb{1} \otimes Z^m)CX = Z^m \otimes Z^m, \tag{8}$$

and by a very similar derivation we obtain

$$CX(\mathbb{1} \otimes Z^{-m})CX^\dagger = Z^m \otimes Z^{-m}. \tag{9}$$

Equations (8) and (9) will prove to be useful later when one needs to decompose unitaries of the form $\exp(i\alpha Z \otimes Z)$. Another useful qudit gate is the *one-level controlled-Z* gate $\overline{CZ}$, which is defined by its action on the basis states $|k\rangle \otimes |l\rangle$ as

$$\overline{CZ}|k\rangle \otimes |l\rangle = \exp\left(i2\pi l\delta_{k,d-1}/d\right)|k\rangle \otimes |l\rangle, \tag{10}$$

where $\delta_{k,l}$ denotes the Kronecker delta. In other words, the gate $\overline{CZ}$ is another generalization of the two-qubit controlled sign flip operation, it applies a phase factor $\exp(i2\pi l/d)$ on the target (second) qudit if the control (first) qudit is in the state $|d-1\rangle$.

## 2.2 Optical multi-rail qudit protocol

One of the simplest way to encode a qudit in an optical system is to use $d$ different optical modes, labeled by the numbers $0, 1, \ldots, d-1$, and map the logical qudit levels to the occupation of different modes via a one-hot encoding map $\|i\rangle\rangle \to |0\rangle_0 |0\rangle_1 \cdots |1\rangle_i \cdots |0\rangle_{d-1}$. This encoding of qudit levels is a multi-rail encoding and has the advantage, that any single-qudit $U(d)$ transformation can be efficiently implemented using the Clements decomposition [29], for illustration see Fig. 1.

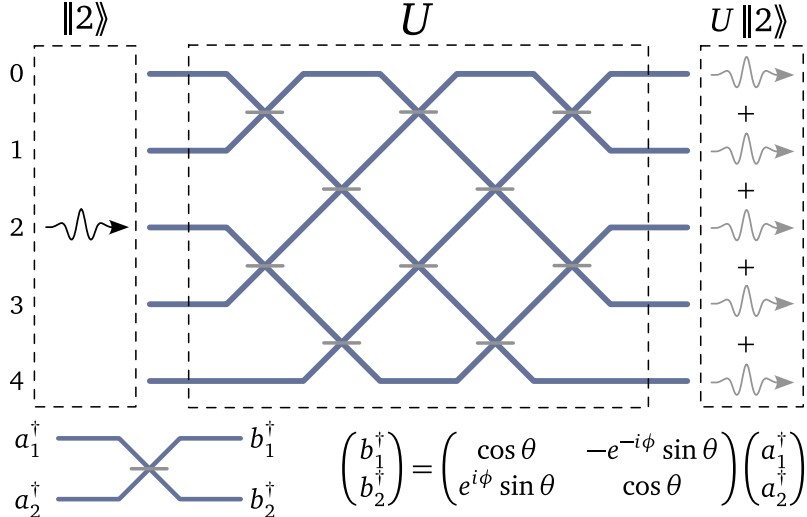

Figure 1: Interferometer configuration for the implementation of a single qudit gate when $d = 5$. This configuration can implement any single qudit gate and minimizes the optical depth. Each crossing represents a general beamsplitter between neighboring optical modes. In general the configuration requires $d(d-1)/2$ number of beamsplitters. On the left the input state $\|2\rangle\rangle$ encoded by single photon enters the interferometer and the output state $U\|2\rangle\rangle$ appears on the right.

When encoding many-qudit systems, one thus needs $d$ optical modes for each qudit, and all the one-qubit operations can be performed by linear optical gates. To implement two-qudit gates, one can generalize the KLM construction, where a nonlinear sign flip operation

is used to obtain non-deterministic two-qubit gates. Moreover, in general not only the sign flip but any phase shift can be implemented this way. The nonlinear phase shift acts on a linear combination of the vacuum, one- and two-photon states of an optical mode, say the $i$-th mode, in the following way $NS_\varphi : \alpha_0 |0\rangle_i + \alpha_1 |1\rangle_i + \alpha_2 |2\rangle_i \rightarrow \alpha_0 |0\rangle_i + \alpha_1 |1\rangle_i + e^{i\varphi}\alpha_2 |2\rangle_i$. It can be implemented in a probabilistic way with linear optics and PNRDs utilizing two ancilla modes. The ancilla modes are prepared in the state $|1\rangle_2 \otimes |0\rangle_3$, so that the initial state of the system looks like $(\alpha_0 |0\rangle_1 + \alpha_1 |1\rangle_1 + \alpha_2 |2\rangle_1) \otimes |1\rangle_2 \otimes |0\rangle_3$. After this preparation of the ancillas, a specific interferometer configuration $U$ is applied on the three modes and then finally the ancillas are measured. If the result of the measurement is $|1\rangle_2 \otimes |0\rangle_3$ then the operation was successful otherwise it failed. There are infinitely many choices for $U$ such that one obtains on the first mode the state $\alpha_0 |0\rangle_1 + \alpha_1 |1\rangle_1 + e^{i\varphi}\alpha_2 |2\rangle_1$ when measuring $|1\rangle_2 \otimes |0\rangle_3$ on the ancilla modes, however, the optimal solution in terms of the probability of success is unique up to some phase factors of the rows and columns of the matrix. The set of solutions for $U$ is the following

$$
U_{11} = 1 - \sqrt{1 - e^{i\varphi}}, \quad |U_{12}| \in \left(0, \sqrt{1 - |U_{11}|^2}\right), \quad U_{13} = \sqrt{1 - |U_{11}|^2 - |U_{12}|^2},
$$

$$
U_{21} = \frac{\sqrt{P}(1 - U_{11})}{U_{12}}, \quad U_{22} = \sqrt{P}, \quad U_{23} = \sqrt{P}\left(\frac{|U_{11}|^2 - U_{11}^* - |U_{12}|^2}{U_{12}U_{13}}\right),
$$

$$
U_{3i} = \sum_{j,k} \epsilon_{ijk} U_{1j}^* U_{2k}^*, \tag{11}
$$

where $\varphi$ is the desired phase shift and

$$
P = \left[\frac{\left|\left(\frac{U_{11}^*(1-U_{11})}{|U_{12}|^2} + 1\right)U_{12}\right|^2}{1 - |U_{11}|^2 - |U_{12}|^2} + \frac{|1 - U_{11}|^2}{|U_{12}|^2} + 1\right]^{-1} \tag{12}
$$

is the probability of success. The optimal solution can be found by simply finding the maximum of $P$ as a function of $|U_{12}|$.

Since all single-qudit gates are already established, it is enough to implement the two-qudit transformation $\overline{CZ}$ defined in Eq. (10) to create a universal gate set [10]. The qudit gate $\overline{CZ}$ can be constructed using beamsplitters and nonlinear phase shift operations, this is shown by Fig. 2. As can be seen the $\overline{CZ}$ gate is made up by $2(d-1)$ number of non-linear phase shifts and the same number of beamsplitters. The nonlinear phase shifts have a success rate less than $1/4$ which is the global maximum of Eq. (12), thus the probability of the $\overline{CZ}$ gate is bound by $P \leq 1/16^{d-1}$. This construction proves that a universal qudit protocol is possible in the same way as in the teleportation-based KLM scheme. However, the proposed $\overline{CZ}$ gate has a very low success rate, which implies that it is an unfeasible option for actual applications (especially when $d$ is high). This means that one needs an alternative method, which we will present in the next subsection.

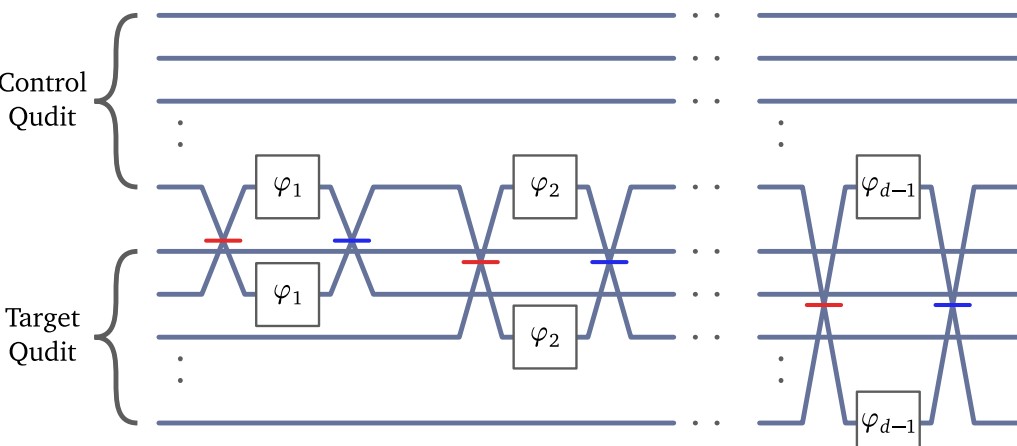

Figure 2: Realization of the $\overline{\text{CZ}}$ gate entangling two qudits defined by the multi-rail encoding. The red and blue segments indicate symmetric beamsplitters with angles $\theta = \pm\pi/4$ and $\phi = 0$ (red is plus and blue is minus), and the white boxes denote nonlinear phase shifts with angles $\varphi_k = 2\pi k/d$.

## 2.3 Locally optimal many-qudit gates

We now present, as an alternative to the scheme of Sec. 2.2, an optimized method for obtaining photonic many-qudit gates, e.g., CZ or $\overline{\text{CZ}}$ operations, with a considerably improved success rate. Similarly to the scheme presented in the previous subsection, the desired gate is executed by implementing a non-trivial unitary transformation and doing postselection on the measurement results, as shown by Fig. 3. The parameters of the unitary are optimized to enhance the success rate. Such a scheme was already employed for generic two-qubits gates successfully [30]. Now we apply the method for generic many-qudit gates.

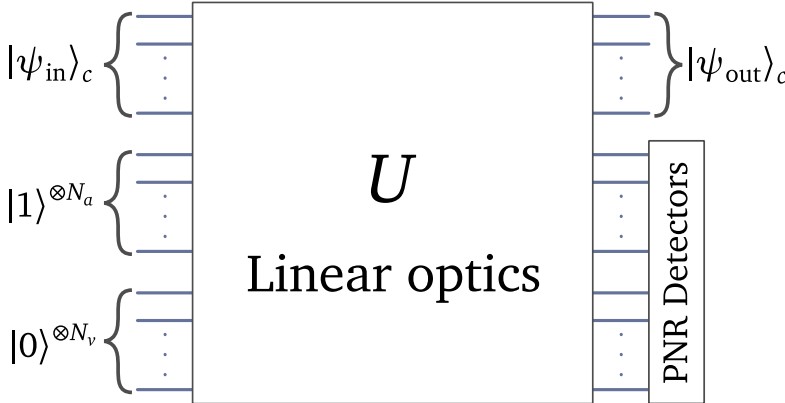

Figure 3: Configuration of the interferometer for a general nonlinear operation. The many-qudit state is encoded in the modes at the top and the ancilla modes are placed below them. The ancilla modes are measured using PNRDs and the measurement result is used to determine whether the operation was successful or not. In the event of success, the computational output is encoded in the many-qudit state $|\psi_{\text{out}}\rangle_c$.

In the multi-rail encoding the many-qudit basis states $\|i_1\rangle\!\rangle \otimes \|i_2\rangle\!\rangle \otimes \cdots \otimes \|i_{N_c}\rangle\!\rangle$ are mapped to photon number states $|n_1, n_2, \ldots, n_M\rangle \equiv |\mathbf{n}\rangle$ with fixed photon number $N_c$ matching the number of qudits and a fixed number of optical modes $M = N_c d$. The general problem we aim to solve is how to implement a many-qudit gate $T$ using linear optics and PNRDs. Given

some computational input state $|\psi_{\text{in}}\rangle_c = \sum_{\mathbf{n}} \alpha_{\mathbf{n}} |\mathbf{n}\rangle$ encoded into the $M$ computational optical modes, the many-qudit gate $T$ transforms it into

$$|\psi_{\text{out}}\rangle_c = T |\psi_{\text{in}}\rangle_c = \sum_{\mathbf{m}} \sum_{\mathbf{n}} T_{\mathbf{mn}} \alpha_{\mathbf{n}} |\mathbf{m}\rangle, \tag{13}$$

where $\mathbf{n}$ and $\mathbf{m}$ indices are referring to the photon number states $|\mathbf{n}\rangle = |n_1, n_2, \ldots, n_M\rangle$ and $|\mathbf{m}\rangle = |m_1, m_2, \ldots, m_M\rangle$ with $\sum_i n_i = \sum_i m_i = N_c$. Unfortunately for a general transformation $T$ there exists no linear interferometer which could execute such a transformation, i.e., most of the time some nonlinear effects are required. Here we will provide nonlinearities via photon number measurements and postselection.

A linear optical interferometer can be described by a unitary $U$ which transforms the creation operators via $a_i'^{\dagger} = \sum_j u_{ij} a_j^{\dagger}$ and the photon number states via

$$U |\mathbf{n}\rangle = \prod_i \left( \sum_j u_{ij} a_j^{\dagger} \right)^{n_i} |0\rangle. \tag{14}$$

Since for most transformations $T$ there exist no $U$ such that $U |\psi_{\text{in}}\rangle_c = |\psi_{\text{out}}\rangle_c$, the next best thing we can achieve is

$$U |\psi_{\text{in}}\rangle_c |1\rangle^{\otimes N_a} |0\rangle^{\otimes N_v} = e^{i\phi} \sqrt{P} |\psi_{\text{out}}\rangle_c |1\rangle^{\otimes N_a} |0\rangle^{\otimes N_v} + \text{terms with other ancilla values}, \tag{15}$$

where we introduced $N_a$ number of ancilla modes each occupied by a single photon and $N_v$ number of vacuum ancilla modes. If we were to prepare the state $U |\psi_{\text{in}}\rangle_c |1\rangle^{\otimes N_a} |0\rangle^{\otimes N_v}$, we would find the ancillas in the state $|1\rangle^{\otimes N_a} |0\rangle^{\otimes N_v}$ with probability $P$. This means that by preparing $U |\psi_{\text{in}}\rangle_c |1\rangle^{\otimes N_a} |0\rangle^{\otimes N_v}$ and measuring the ancilla modes $T$ is implemented with probability $P$. The relation between $U$ and $T$ is

$$\langle \tilde{\mathbf{m}}| U |\tilde{\mathbf{n}}\rangle = \tilde{U}_{\mathbf{mn}} = e^{i\phi} \sqrt{P} T_{\mathbf{mn}}, \tag{16}$$

where $|\tilde{\mathbf{n}}\rangle = |\mathbf{n}\rangle |1\rangle^{\otimes N_a} |0\rangle^{\otimes N_v}$. The Fock space matrix elements $\tilde{U}_{\mathbf{mn}}$ can be expressed in terms of $u_{ij}$ using permanents [31]. Equation (16) is a nonlinear polynomial equation for $u_{ij}$ and it can be solved for $u_{ij}$ using numerical optimization methods while maximizing the success rate $P$. Since $u$ is unitary it also has to satisfy $uu^{\dagger} = \mathbb{1}$. To solve this problem via optimization we can define the cost function

$$L = F + \lambda P + \sigma C, \tag{17}$$

where

$$F = \frac{\left| \text{Tr}\left( \tilde{U}^{\dagger} T \right) \right|^2}{\text{Tr}\left( \tilde{U}^{\dagger} \tilde{U} \right) \text{Tr}\left( T^{\dagger} T \right)}, \qquad P = \frac{\text{Tr}\left( \tilde{U}^{\dagger} \tilde{U} \right)}{\text{Tr}\left( T^{\dagger} T \right)}, \qquad C = -\text{Tr}\left\{ \left( uu^{\dagger} - \mathbb{1} \right)^2 \right\}, \tag{18}$$

and $\lambda$ and $\sigma$ are positive real constants. The fidelity $F$ reaches its maximum if and only if Eq. (16) is satisfied for some probability $P$, this is guaranteed by the Cauchy–Schwarz inequality. The unitary constraint $C$ reaches its maximum if and only if $U$ is a unitary matrix.

The optimization problem defined by $L$ was solved using a trust-region method [32]. The optimization process went as follows. First the problem was solved using $\lambda = 0$. Once a solution with $F = 1$ and $C = 0$ was found $\lambda$ was gradually increased to the largest possible value for which the solution still converged to $F = 1$. At the beginning $N_a$ and $N_v$ were chosen such that the system of equations in Eq. (16) would be under-defined. After finding a solution $N_a$ and $N_v$ would be decreased until no solution with $F = 1$ and $C = 0$ could be found. This process was carried out for the qutrit $\overline{CZ}$ and $CZ$ gates (see Table 1). There is a significant improvement in the success rate when the gates are optimized. These optimized gates could be

used to non-deterministically prepare entangled photon states which are necessary for many measurement based architectures like optical cluster state computing [33]. In the following sections we will look at how high-dimensional cluster states might be used in photonic qudit schemes in general and in particular for the $k$-coloring problem.

| $T$ | $P_{\text{naive}}$ | $P$ | $N_a$ | $N_v$ |
|---|---|---|---|---|
| CZ (qubit) [34] | 0.0625 | $0.0740\ldots$ | 2 | 0 |
| $\overline{\text{CZ}}$ (qutrit) | $\approx 0.0016$ | 0.011 | 3 | 3 |
| CZ (qutrit) | $\approx 0.00000256$ | 0.000507 | 5 | 4 |

Table 1: Optimized two-qutrit gates. $P_{\text{naive}}$ is the probability we get when the gates are decomposed into many nonlinear phase shifts. $P$ is the best probability found after doing the optimization process for many different randomly selected initial conditions.

# 3    High-dimensional cluster states

The most common computational model is the quantum circuit model which resembles the logic gate based description of classical computation. Measurement based quantum computation is an alternative to the circuit model, where instead of applying unitary gates one performs adaptive single qubit or qudit measurements on a large entangled state, called the cluster state. At first this may seem completely different from the circuit model but their equivalence have been proved [28]. MBQC is in particular a promising paradigm for the realization of universal optical quantum computation, because if cluster states can be prepared efficiently there is no longer need for further non-deterministic gates to perform the computation. In other words, MBQC shifts the problem of implementing nonlinearities to the problem of state preparation. Several methods have been developed to generate cluster states efficiently, thus MBQC is one of the most viable options for realizing universal quantum computation using photonics [35–37].

Cluster states have the form

$$\prod_{\{i,j\}\in C} CZ_{i,j} \, |+\rangle^{\otimes N} , \tag{19}$$

where $CZ$ is the controlled qudit $Z$ gate and $C$ is a graph whose edges represent the $CZ$ gates between qudits. The idea is to perform single-qudit measurements sequentially on the qudits found in the cluster, where the basis of each measurement depends on the results of the previous measurements. When a qudit is measured it is effectively removed from the cluster reducing its size. After the measurements, the remains of the cluster state will encode the computational state up to some known single-qudit Pauli errors which can be easily corrected for at the end of the computation.

To connect the cluster state picture with the quantum circuit model, we can use the teleportation identity

$$
\begin{array}{c}
|\psi\rangle \;\rule{0pt}{0pt}\!\!\bullet\!\!\rule{0pt}{0pt}\; \boxed{P(\boldsymbol{\theta})} \; \boxed{H^{\dagger}} \; \boxed{\nearrow} \quad = m \\[4pt]
|+\rangle \;\boxed{Z}\;\rule{0pt}{0pt}\hspace{4cm} X^{m}HP(\boldsymbol{\theta})|\psi\rangle
\end{array}
\tag{20}
$$

where $|\psi\rangle$ is a general qudit state, $m$ is the result of the measurement performed on the top qudit, and $P(\boldsymbol{\theta})$ is a phase gate with $\boldsymbol{\theta}\in\mathbb{R}^d$ which acts like $P(\boldsymbol{\theta})|n\rangle = \exp(i\theta_n)|n\rangle$ [38]. This circuit represents a single measurement step in the cluster state picture. It applies the single-qudit unitary $HP(\boldsymbol{\theta})$ and propagates the state to the qudit that connects to it. Any unitary

can be decomposed into a product that only contains gates of the form $HP(\boldsymbol{\theta})$, thus chaining together many of these circuits one can perform any single-qudit unitary [38, 39]. These chains directly correspond to measurements on a linear cluster state. An important detail is that after each measurement we introduce Pauli errors of the form $X^a Z^b$ where $a, b$ depend on the results of the previous measurements. In order to account for the accumulated Pauli errors we have to change the measurement angles to $\boldsymbol{\theta}'$ such that $HP(\boldsymbol{\theta}')X^a Z^b = X^{a'} Z^{b'} HP(\boldsymbol{\theta})$.

In general, we can translate any quantum circuit to a cluster state with measurement instructions using the teleportation identity defined above. This way each qudit in the quantum circuit model will correspond to a linear subcluster made up of several physical qudits and the $CZ$ gates between qudits in the quantum circuit will become connections across the corresponding linear clusters. Although this construction is enough to translate any quantum circuit, it makes the translation easier if we allow the use of $CZ^\dagger$ connections between physical qudits in the cluster. This is justified by another teleportation identity slightly different from (20)

$$
\begin{array}{c}
|\psi\rangle \longrightarrow \bullet \boxed{P(\boldsymbol{\theta})} \boxed{H} \boxed{\nearrow} = m \\
|+\rangle \boxed{Z^\dagger} \underline{\hspace{3cm}} X^m H^\dagger P(\boldsymbol{\theta})|\psi\rangle
\end{array} \qquad . \tag{21}
$$

These cluster states can be illustrated by depicting the linear clusters as rows with the $CZ$ or $CZ^\dagger$ gates between them shown as vertical lines. An example of this is shown by Fig. 4.

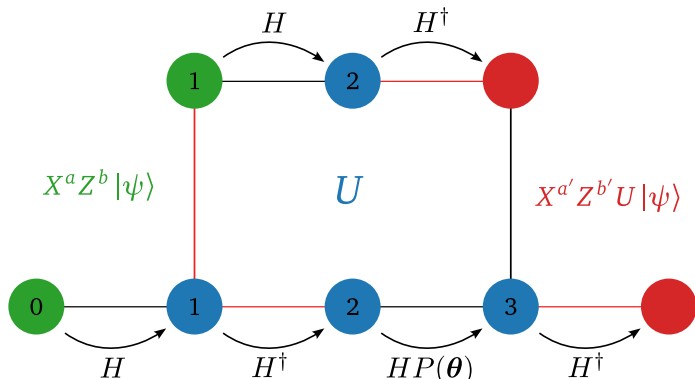

Figure 4: Qudit cluster state with the measurement instructions to implement the two qudit unitary $U = CXP(\boldsymbol{\theta})CX^\dagger$. The black and red edges denote the $CZ$ and $CZ^\dagger$ connections between the physical qudits. The initial state with its Pauli errors is encoded in the green qudits. The measurement order is indicated by the numbers. After the execution of all the measurements, the final state is encoded in the red qubits. The arrows show how the qudit states are teleported from one qudit to the next, and the unitary of each arrow indicates what transformation should the teleportation perform.

## 4 The $k$-coloring problem

Graph coloring is one of the most studied topics of graph theory because of its applications in solving scheduling problems and compiler theory [40, 41]. Coloring refers to the assignment of labels (colors) from a label set to the vertices of a graph $G$. This assignment defines a map $C : V \to L$, where $V$ is the set containing the vertices of $G$ and $L$ is the label set. Arbitrarily ordering the vertices of $G$ allows us to represent a coloring as string of labels: If we call the $i$-th

vertex $v_i \in V$ and its assigned color $C(v_i) = l_i \in L$, then the string $\mathbf{l} = l_1 l_2 l_3 \ldots l_{|V|}$ completely characterizes the coloring $C$. The $k$-coloring problem of a graph is to decide whether the vertices of the graph can be colored using $k$ colors, such that no adjacent vertices have the same color. The smallest $k$ for which the $k$-coloring is possible for a given graph is called the chromatic number of the graph. The $k$-coloring problem of an arbitrary graph is known to be NP-complete [42–44]. There has been rigorous proof showing the speed up of certain quantum algorithms compared to the best known classical, [45], and even certain heuristic variational quantum algorithms for a similar problem have been hinted to outperform classical methods [46].

## 4.1 Qubit algorithm

The $k$-coloring problem can be reformulated as a combinatorial optimization problem where the goal is to minimize the number of edges connecting vertices with identical colors by searching through the possible $|V|$ length strings of the label set $L = \{0, 1, 2, \ldots, k-1\}$. On qubit-based architectures, binary optimization problems can be solved using the quantum approximate optimization algorithm (QAOA) [47]. Encoding the strings of $L$ as bit-strings means that we can use the QAOA to solve the $k$-coloring problem as well; such a QAOA type of approach for the $k$-coloring (and the related $k$-SAT problem) has been studied in Refs. [48–52]. The encoding of colorings into bit-strings is most easily done via one-hot encoding, this means that the strings of $L$ are encoded into $N = |V| \cdot k$ bits, where each set of $k$ consecutive bits encodes the color of a vertex. The color of the $i$-th vertex $l_i$ is indicated by the $i$-th set of $k$ bits of the form $00\ldots010\ldots00$, where the $l_i$-th bit is flipped and the rest of the $k$ bits is zero. We will denote the $j$-th bit of the $i$-th set of $k$ bits by $x_{i,j}$.

Minimizing the cost function $f : \{0,1\}^N \to \mathbb{R}$

$$f(\mathbf{x}) = C \sum_{n=1}^{|V|} \left(1 - \sum_{i=0}^{k-1} x_{n,i}\right)^2 + D \sum_{n=1}^{|V|} \sum_{m=1}^{|V|} \sum_{i=0}^{k-1} A_{nm} x_{n,i} x_{m,i}, \tag{22}$$

is equivalent to finding a solution to the $k$-coloring problem, where $\mathbf{x}$ is a bit-string of length $N$, $A_{nm}$ are the matrix elements of the adjacency matrix of $G$ and $C$ and $D$ are arbitrary positive real constants. The first term in Eq. (22) weighted by $C$ is a penalty term, since not every bit-string of length $N$ corresponds to a coloring. For every $n \in \{1, 2, \ldots, |V|\}$ exactly one of the $k$ bits $x_{n,j}$, where $j \in \{0, 1, \ldots, k-1\}$, has to equal one in order for $\mathbf{x}$ to encode a coloring. The remaining term simply counts the number of edges that connect vertices with the same color. To construct the QAOA cost Hamiltonian we can use the recipe

$$H_C = \sum_{\mathbf{x} \in \{0,1\}^N} f(\mathbf{x}) |\mathbf{x}\rangle \langle \mathbf{x}|, \tag{23}$$

where $|\mathbf{x}\rangle$ is the computational state

$$\bigotimes_{n=1}^{|V|} \left( \bigotimes_{i=0}^{k-1} |x_{n,i}\rangle \right).$$

Substituting Eq. (22) into Eq. (23), the cost Hamiltonian becomes

$$H_C = \frac{C}{4} \sum_{n=1}^{|V|} \left( 2\mathbb{1} - \sum_{i=0}^{k-1} (\mathbb{1} - Z_{n,i}) \right)^2 + \frac{D}{4} \sum_{n=1}^{|V|} \sum_{m=1}^{|V|} \sum_{i=0}^{k-1} A_{nm} (\mathbb{1} - Z_{n,i})(\mathbb{1} - Z_{m,i}), \tag{24}$$

where $Z_{n,i}$ is the Pauli Z matrix acting on the $(k \cdot (n-1) + i)$-th qubit labeled by the tuple $(n, i)$. Obtaining the ground state of $H_C$ is the same as finding a bit-string that minimizes the cost function $f$ [49].

QAOA minimizes the expectation value $\langle\psi(\boldsymbol{\alpha},\boldsymbol{\beta})|H_C|\psi(\boldsymbol{\alpha},\boldsymbol{\beta})\rangle$, where

$$|\psi(\boldsymbol{\alpha},\boldsymbol{\beta})\rangle = \left(\prod_{n=1}^{p} e^{i\alpha_n H_M} e^{i\beta_n H_C}\right)|+\rangle^{\otimes N}, \tag{25}$$

$\boldsymbol{\alpha},\boldsymbol{\beta}\in\mathbb{R}^p$ and

$$H_M = \sum_{n=1}^{|V|}\sum_{i=0}^{k-1} X_{n,i}$$

is the mixing Hamiltonian, where $p$ denotes the number of QAOA layers. Preparation of $|\psi(\boldsymbol{\alpha},\boldsymbol{\beta})\rangle$ requires the implementation of two parameterized unitary $U_M(\alpha) = e^{i\alpha H_M}$ and $U_C(\beta) = e^{i\beta H_C}$. $U_M$ can be realized using single qubit rotations and only $U_C$ requires entangling gates. $U_C$ can be further separated into the product of two unitaries, where one of the unitary requires only single qubit phase gates and the other unitary requires only controlled rotations. This can be achieved by separating the cost Hamiltonian into a non-interacting and an interacting part, $H_C = H_0 + H_1$, where

$$H_1 = \frac{C}{2}\sum_{n=1}^{|V|}\sum_{i<j=0}^{k-1} Z_{n,i}Z_{n,j} + \frac{D}{2}\sum_{\{n,m\}\in E}\sum_{i=0}^{k-1} Z_{n,i}Z_{m,i}. \tag{26}$$

In Eq. (26) $E$ denotes the set containing the edges of the graph. The Hamiltonians $H_0$ and $H_1$ commute since they only contain Pauli Z matrices, thus

$$U_C(\alpha) = e^{i\alpha H_0} e^{i\alpha H_1}. \tag{27}$$

The unitary $e^{i\alpha H_1}$ can be decomposed into the product of $|V|\binom{k}{2} + k|E|$ number of controlled rotations

$$e^{i\alpha H_1} = \prod_{n=1}^{|V|}\left(\prod_{i<j=1}^{k} CX_{n,i;n,j} P_{n,j}(\alpha C/2) CX_{n,i;n,j}\right) \times \prod_{\{n,m\}\in E}\left(\prod_{i=0}^{k-1} CX_{n,i;m,i} P_{m,i}(\alpha D/2) CX_{n,i;m,i}\right), \tag{28}$$

where $P_{n,j}(\alpha) = e^{i\alpha Z_{n,j}}$ is the phase gate acting on the qubit labeled by the pair of indices $(n,j)$ and $CX_{n,i;n,j}$ is a controlled NOT gate between the control qubit $(n,i)$ and the target qubit $(n,j)$. Eq. (28) can be derived using the identity $e^{i\alpha Z\otimes Z} = CX(\mathbb{1}\otimes P(\alpha))CX$.

The physical realization of the unitary $e^{i\alpha H_1}$ requires by far the most amount of physical resources compared to the other parts of the algorithm, since only this part needs entangling gates. Therefore, focusing only on the implementation details of $e^{i\alpha H_1}$ can give us a good lower bound on the resources necessary to carry out the QAOA.

## 4.2 Graph coloring using qudits

When using qudits with dimension $d = k$, there is a one-to-one correspondence between the computational basis and the possible graph colorings with $k$ colors via the mapping

$$l_1 l_2 l_3 \ldots l_{|V|} \rightarrow |l_1\rangle \otimes |l_2\rangle \otimes |l_3\rangle \otimes \cdots \otimes |l_{|V|}\rangle, \tag{29}$$

where $l_i \in \{0,1,2,\ldots,k-1\}$. This means that the coloring problem can be formulated as an unconstrained $d$-ary optimization problem which can be solved by the QAOA generalized to qudits [53–55]. Because the optimization problem is unconstrained, the decomposition of the QAOA layers will turn out to be simpler due to lack of penalty terms. The QAOA for qudits is very similar to the qubit version only the form of the mixing and cost Hamiltonian is different.

The mixing Hamiltonian is still a simple sum of single-qudit terms, e.g., the $r$-nearby-values single-qudit mixer $H_M = \sum_{n=1}^{|V|} \sum_{i=1}^{r} \left( X_n^i + X_n^{\dagger i} \right)$ as in Ref. [54], and the cost Hamiltonian is

$$H_C = \sum_{n=1}^{|V|} \sum_{m=1}^{|V|} \sum_{i=0}^{k-1} A_{nm} Z_n^i Z_m^{k-i} = \sum_{\{n,m\} \in E} \sum_{i=0}^{k-1} Z_n^i Z_m^{k-i}, \tag{30}$$

where $Z_n$ is the qudit Pauli $Z$ gate acting on the $n$-th qudit and $A_{nm}$ are the matrix elements of the adjacency matrix of the graph $G$. The ground states of the Hamiltonian $H_C$ correspond to optimal $k$-colorings of $G$. Similarly to the qubit case, we can decompose the unitary $U_C(\alpha) = e^{i\alpha H_C}$ into $|E|$ number of controlled qudit rotations

$$e^{i\alpha H_C} = \prod_{\{n,m\} \in E} CX_{n;m} P_m(\alpha) CX_{n;m}^{\dagger}, \tag{31}$$

where $P_m(\alpha)$ is a single-qudit phase gate which acts on the $m$-th qudit the following way

$$P_m(\alpha) |\mathbf{x}\rangle = e^{i\alpha \sum_{n=0}^{k-1} Z_m^n} |\mathbf{x}\rangle = \begin{cases} e^{i\alpha k} |\mathbf{x}\rangle & \text{if } x_m = 0, \\ |\mathbf{x}\rangle & \text{otherwise.} \end{cases} \tag{32}$$

### 4.3 Cluster states for the qubit and qudit implementations

Having established all the necessary techniques to implement the QAOA solution of the $k$-coloring problem both using qubits and qudits, we can now compare the physical resources used by the two methods. A single layer of the qudit QAOA algorithm without the mixing part can be executed on the qudit cluster state $C_d$ constructed from many copies of $R_d$, where $R_d$ is a cluster with the topology depicted by Fig. 4 made up by $d$-dimensional qudits. When the number of colors is $k$, one has to use $d=k$-dimensional qudits for the algorithm. The construction of $C_d$ is shown by Fig. 5. The construction of $C_2$, the qubit equivalent of $C_d$, works differently. In the qubit case, one has to add $k$ copies of $R_2$ for each edge of $G$ and also $k(k-1)/2$ copies for each node of $G$. These construction rules follow from the decompositions in Eqs. (28) and (31).

The number of qudits in $C_d$ is approximately

$$|C_d| \approx |R_d| \cdot |E| = 8|E|, \tag{33}$$

where $|E|$ is the number of edges in $G$, which is equal to the number of copies of $R_d$ used to build $C_d$, and $|R_d|$ is the number of qudits in $R_d$. The approximation can be made exact by carefully considering the precise sequence of node additions and removals in the cluster growing process, however, for large enough graphs Eq. (33) provides a highly accurate estimate. Similarly, the number of qubits in $C_2$ is

$$|C_2| \approx |R_2| \cdot \left( \binom{k}{2} |V| + k|E| \right) = 8 \binom{k}{2} |V| + 8k|E|. \tag{34}$$

In general, the ratio $|C_2|/|C_d|$ is strictly larger than $k$, and in the large dense graph limit, when $|E| \gg |V|$, this ratio approaches $k$. This is illustrated by Fig. 6 for Erdős-Rényi random graphs. If one uses the multi-rail encoding described in Sec. 2, the photon numbers of the cluster states is equal to their size. The number of optical modes required to encode the cluster state $C_d$ is $d|C_d|$ including the qubit case $d = 2$. Thus, the number of photons decreases by a $k$-fold and the number of optical modes halves, when using multi-rail qudits instead of KLM qubits to encode the $k$-coloring problem.

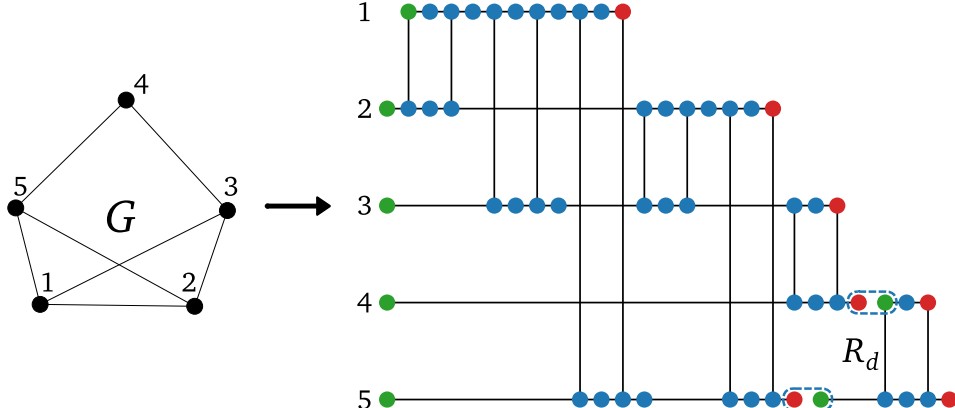

Figure 5: Qudit cluster state for a single layer of the qudit QAOA. Each edge of $G$ corresponds to a controlled qudit rotation in the decomposition of $\exp(i\alpha H_C)$ described by Eq. (31). Thus one can chain many copies of $R_d$ to form the complete qudit cluster. The two input nodes of $R_d$ are merged with two output nodes of the unfinished cluster state based on the topology of $G$. Each row of physical qudits represents a logical qudit labeled by the numbers. Each logical qudit encodes the color of a node in $G$.

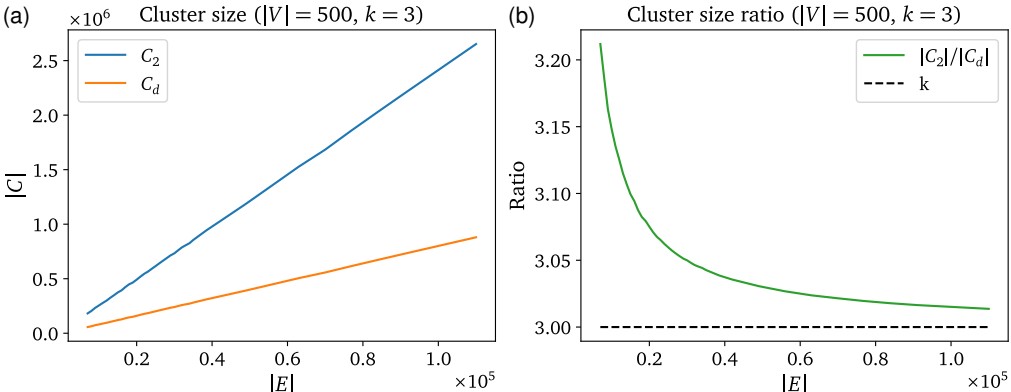

Figure 6: (a) Number of physical qubits and qudits versus the number of edges in $G$, the graph to be colored. $|C_2|$ is the number of physical qubits in the qubit cluster state implementation of $\exp(i\alpha H_1)$, where $H_1$ is defined by Eq. (26). $|C_d|$ is the number of physical qudits in the cluster state implementation of $\exp(i\alpha H_C)$, where $H_C$ is defined in Eq. (30). (b) Ratio of the qubit and qudit cluster sizes when $|V| = 500$ and $k = 3$, where $|V|$ is the number of vertices in $G$ and $k$ is the number of colors used for the coloring.

## 5   Conclusion and outlook

In this work, we investigated a KLM-like qudit encoding for universal photonic quantum computation. First, the optimization of non-deterministic many-qudit gates was discussed, then the solution of the $k$-coloring problem as an application of high-dimensional cluster states was examined.

The physical implementation of the single-qudit gates by the described protocol should pose no problem in experiments, since they require only the use of passive linear optics. How-

ever, the low probabilities of the entangling many-qudit gates make this scheme applied with postselection challenging from the universal quantum information processing point of view. Nevertheless, the approach may still be used to non-deterministically prepare highly entangled photon states, which are required by many other universal photonic schemes, most notably photonic cluster state computation. We present a numerical optimization method to improve the success rate of a general many-qudit gate. The method is capable of finding locally optimal solutions with perfect fidelity, improving the success rates typically by orders of magnitude compared to the naive implementations when the gates are decomposed into nonlinear phase shift operations. We only considered qudit gates where photon number of the input states is fixed, but in principle it could also be used when this restriction is lifted. For example, it can be applied to numerically find an optimal solution for the nonlinear phase shift. Furthermore, it can be employed for more general hybrid qudit gates [56], when the dimensions of the qudits are not the same.

To amend the problems of the non-deterministic postselection based approach, we used cluster states encoded in the multi-rail encoding to give an implementation for the high-dimensional QAOA. We demonstrated our method on the $k$-coloring problem which can be solved by either using qubits or qudits with $d=k$. We show that the high-dimensional cluster states proposed in this work to solve the $k$-coloring problem are made up of fewer photons and require less optical modes to encode than the KLM qubit cluster states needed to perform the same task. The reduced number of photons can help with quantum storage since usually the fidelity of quantum memories drop quickly with number of photons due to photon loss [57]. The real difficulty of cluster state computation is in the generation of the cluster states, and all results involving cluster states rely on the assumption that they can be generated efficiently. In a recent work it has been shown that in theory one can produce these type of high-dimensional cluster states deterministically using quantum emitters [58]. The methods shown in Ref. [58] combined with the result of this work give a convincing argument for the usefulness of multi-rail encoded high-dimensional cluster states.

# Acknowledgements

**Funding information**   MK and LO was supported by the Ministry of Culture and Innovation and the National Research, Development and Innovation Office within the Quantum Information National Laboratory of Hungary (Grant No. 2022-2.1.1-NL-2022-00004) and grants K131938, K142179, FK 135220. ZZ was supported supported by TKP Project no. TKP2021-NVA-04 financed under the TKP2021-NVA funding scheme. LO also acknowledges support from of the NRDI Office of Hungary and the Hungarian Academy of Sciences through the Bolyai and Bolyai+ scholarships.

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
