# Peer review of "Efficient qudit based scheme for photonic quantum computing"

_SciPost Physics Core_

## Round 2 · Author Response

Dear Editor,

We hereby resubmit our manuscript for publication in SciPost Physics Core. The previous version was reviewed a referee with an overall positive assessment of our work. The referee also made a couple of recommendations and requested changes. With the aim to further improve our manuscript we have prepared a new and extended version of the manuscript, in this all of the Referee's comments are addressed. We detail the requested list of changes of the Referee and how we addressed thesel in the "List of changes" box.

On behalf of all authors,
Zoltán Zimborás

---

## Round 2 · List of Changes

List of requested changes:

1 - "A main motivation for realising cluster states seems to be that they require less optical modes and less entangling states than qubit cluster states with analogous computational capabilities. Although it is clear how this can be advantageous, it seems it could come at the expense of experimental realisability. The authors may want to cite the relevant experimental literature comparing the feasibility of the two."

In the new version of the manuscript, we cited in the introduction (5th paragraphs) several papers from the literature (Refs. [15, 29-33]) that show the possible advantages of qudit cluster states and their experimental feasibility:

"For the above mentioned reasons there has been a considerable effort to realize qudit entanglement experimentally [15, 29-33] and demonstrate the benefits of high-dimensional quantum information processing. Observing the experimental setups used in these experiments shows that there is a clear trade off between the experimental complexity and the high-dimensionality of the experiment. However, these experiments also show that the increased complexity can be overcome using clever engineering and it is possible to demonstrate advantages of high-dimensional quantum information processing experimentally."

2- "In the beginning of Section 2.2 the authors present the multi-rail qudit encoding as the simplest way to encode a qudit in an optical system. I would suggest including references supporting this statement and a brief discussion stating the disadvantages of the alternatives."

We added a new paragraph on Section 2.2 detailing the advantages and disadvantages of multi-rail encoding and of its the most common alternative, the frequency comb qudits:

"Another competing encoding for discrete-variable photonic quantum computation are frequency comb qudits. In concept they are very similar to multi-rail qudits, but instead of spatial modes the qudit levels are represented by different frequencies. One of the advantages of this frequency encoding is that the quantum information can be transported in a single fiber-optical cable over long distances [35]. However, creating interference between modes with different frequencies requires either $\chi^2$, $\chi^3$ nonlinearities or electro-optic modulation (EOM). Such nonlinearities have not yet been proven to be scalable to high-dimensional Hilbert spaces, and although EOMs can be cascaded in a scalable manner to reproduce any operation possible with linear-optics on spatially encoded qudits, the realization of EOMs with arbitrary modulation patterns pose significant practical challenges compared to the challenges of realizing linear-optical networks [36]. For this reason, spatially encoded multi-rail qudit registers are currently experimentally more viable than frequency comb qudit registers, as large programmable linear-optical networks have been built and efficient single-photon generation capabilities have been developed [37]."

3- "In the optimisation protocol the authors dynamically modify the optimal point by modulating the different parameters of the cost function. I wonder if they could explain in detail the reasoning behind this choice and what made them prefer the trust-region method to all other alternatives that could be implemented in a more direct manner. Moreover, is the optimum found by the authors guaranteed to be the global optimum? "

- We clarified that the solution found are not global optima (see Table 2). In Section 2.3.

- In the beginning of paragraph 4 of Section 2.3, we explained why we chose a trust region method:

"The optimization problem defined by the cost function L was solved using a trust-region method [40] which, beside direct line search methods [41], is one of the most established family of algorithms for solving non-linear unconstrained optimization problems. We chose this technique because it has been shown that trust region methods in certain cases can have better convergence properties than direct line search methods [40], and the implementations of the trust-region methods provided by the Scipy Python package [42] proved adequate."

- At the end of paragraph 4 of Section 2.3, we added an explanation why we modulated the the different parameters of the cost function:

"This process proved practical for two reasons. The first reason was that the local optimum to which the algorithm converged to from a random staring point did not necessarily lie on the F = 1 and C = 0 surface. By performing the optimization process with $\lambda$ set to zero, we checked that the algorithm can actually reach the F = 1 and C = 0 surface. This initial checking run saved time as the optimization process could be restarted from a new initial point when it was clear that the solution will not reach the F = 1 and C = 0 surface. The second reason was that if the optimization run succeeded with $\lambda = 0$, i.e., the solution reached the F = 1 and C = 0 surface, the solution could be used as a starting point for the complete optimization problem when $\lambda > 0$."

4 - "The optimisation performed by the authors is justified in light of the acceptance probability of the CZ gate scaling ad 1/16^{d-1} which it unusable for practical applications. The authors then proceed to provide their alternative scheme but do not provide a lower bound above which the gate could be considered “feasable” or “practically useful”. This would help benchmark the actual impact provided by the optimisation procedure."

In the last 3 paragraphs of Section 2.3, we discussed the feasibility of the optimised and unoptimised two-qudit gates in the light of the currently available technology:

"To further discuss the impact of these results it is useful to make an estimate of what is practically achievable in the near term using already existing single-photon generation and detection devices. Highly indistinguishable single photons can be generated on-demand using quantum dots [45-47]; state-of-the-art sources can generate as much as 100 million single photons per second [37] potentially allowing for MHz attempt rates when implementing non-deterministic entangling gates such as the ones listed in Table 1. When it comes to the detection of single photons, superconducting nanowire single-photon detectors (SNSPDs) are the leading technology in the near-infrared region [48-51]. The advantages of SNSPDs are that they have exceptionally low dark count rates, high system detection efficiency (SDE) and low reset times on the order of nanoseconds. These properties make SNSPDs ideal for quantum information processing applications.

The increased success probabilities of the optimized non-deterministic gates presented in this section provide a speedup over their naive variants. The speed of the non-deterministic gate synthesis is directly proportional to the success probability of the gate and the rate of attempts which is dictated by the reset time of the detectors and the emission rate of single-photon sources. In previous boson sampling experiments single-photon pulse trains generated by quantum dots were demultiplexed and injected into several spatial modes at rates close to 1 MHz [52,53]. An experimental setup similar to what is presented in Refs. [52,53] could be used to implement non-deterministic gates with an attempt rate in the MHz range. At such a rate even the naive variants of the gates listed in Table 1 would be technically feasible under ideal circumstances. For example, the naive non-deterministic qutrit (d = 3) CZ gate would take roughly 4*10^5 attempts on average to succeed, i.e., 0.4 seconds if 10^6 attempts are made every second. On the other hand, the optimized qutrit CZ gate would take only 2 milliseconds on average to succeed under the same circumstances.

While the naive gates are feasible by themselves, in practice there can be strict timing constraints on the gate times in quantum information processing schemes which require quantum memories. In many cases the storage time of quantum memories poses as a limit on the gates times, e.g., in the KLM scheme or in cluster state preparation schemes like Ref. [54]. For these and similar applications the gate times should be much shorter than the storage time of the quantum memories. Current quantum memories operating on the single-photon level typically have storage times between a few milliseconds to a second at most [55,56]. Thus, in the near term it is unlikely that non-deterministic gates with average gate times longer than a few milliseconds would be practical for universal quantum computing schemes. Also, the efficiency of quantum memories usually decreases exponentially with the storage time (see Refs. [55,56]), further emphasizing the need for speed optimized gates. The optimized gates in Table \ref{tab:optimization} show an improvement over the success probabilities of the naive gates and hence reduces the average gate times by orders of magnitude. For the reasons above, these improvements can have a significant impact on the feasibility of experimental realization of high-dimensional universal quantum computing schemes."

5-"From Table I, where the authors report the naive and optimised acceptance probabilities for the two-qutrit gates, it is clear that there is an advantage provided by the optimisation procedure. The authors should however show how this advantage scales with d, to make sure it is not only a different prerefactor in and analogous unfavourable scaling."

- In Table 1 we added an extra entry for the CZ-bar two-audit gate (in the d=4 case).

- We added Figure 4 to compare the scaling of the success probabilities of optimised and unoptimised CZ-bar gates with the dimension d.

- In paragraph 5 of Section 2.3, we added a discussion on the scaling of success probabilities with the dimension d.

---

## Editorial Decision

accepted_in_target_journal